# High-Efficiency Oxygen Reduction Reaction Revived from Walnut Shell

**DOI:** 10.3390/molecules28052072

**Published:** 2023-02-22

**Authors:** Lei Yan, Yuchen Liu, Junhua Hou

**Affiliations:** 1School of Physics and Information Engineering, Shanxi Normal University, No. 339 Taiyu Road, Xiaodian District, Taiyuan 030031, China; 2Extreme Optical Collaborative Innovation Center, Shanxi University, No. 92, Wucheng Road, Xiaodian District, Taiyuan 030006, China; 3Modern College of Humanities and Sciences, Shanxi Normal University, No. 501 Binhe West Road, Yaodu District, Linfen 041000, China

**Keywords:** biomass, electrocatalysis, nitrogen doping, urea N-doped

## Abstract

The development of inexpensive and efficient electrocatalysts for oxygen reduction reactions (ORR) remains a challenge with respect to renewable energy technologies. In this research, a porous, nitrogen-doped ORR catalyst is prepared using the hydrothermal method and pyrolysis with walnut shell as a biomass precursor and urea as a nitrogen source. Unlike past research, in this study, urea is not directly doped; instead, a new type of doping is carried out after annealing at 550 °C. In addition, the sample’s morphology and structure are analyzed and characterized by scanning electron microscopy (SEM) and X-ray powder diffraction (XRD). A CHI 760E electrochemical workstation is used to test NSCL-900’s performance in terms of oxygen reduction electrocatalysis (ORR). It has been found that the catalytic performance of NSCL-900 is significantly improved compared with that of NS-900 without urea doping. In a 0.1 mol/L KOH electrolyte, the half-wave potential can reach 0.86 V (vs. RHE) and the initial potential is 1.00 V (vs. RHE). The catalytic process is close to four-electron transfer and there are large quantities of pyridine nitrogen and pyrrole nitrogen.

## 1. Introduction

With the uncontrolled exploitation of fossil fuels, environmental problems are becoming increasingly serious [1]. The development of conversion technology to produce renewable energy or clean energy is a significant way to achieve green [2], economic, and sustainable development [3]. In hydrogen fuel cells, cathode oxygen reduction occurs slowly, so it requires the use of catalysts, among which the most effective catalysts for ORR are Pt-based noble metal catalysts (such as P_t_/C). However, the use of Pt-based catalysts has been limited in large-scale commercial applications due to their high cost [4], low Earth reserves, long-term instability, and inactivation [5]. Therefore [6], the development of non-noble metal catalysts and non-metal catalysts has received widespread attention from researchers. So far, different kinds of non-noble metal electrocatalysts have been explored, including metal nitrogen-doped carbon materials, transition metal oxides, carbides and nitrides, and non-metal heteroatom-doped carbon-based materials in the above electrocatalysts. Metal-free heteroatom-doped carbon materials are among the most promising ORR materials due to their low cost, controllable pore structure, high electron conductivity, and high tolerance to impurities. Among them [7], non-metal catalysts are widely regarded as the most representative catalysts in terms of replacing Pt-based catalysts [8].

Biomass waste is rich in carbon atoms [9]. The carbon nanostructures formed by high-temperature carbonization and activation have well-developed internal pores [10], high surface area, high conductivity, and corrosion resistance [10]. As an efficient carrier catalyst for oxygen reduction [11], bio-derived carbon can uniformly combine with transition metals and nonmetal atoms to generate many active sites, thus rendering it an ideal material for fuel cell catalysts [12]. Many types of biomass waste are used to produce carbon, such as walnut skin, human hair, awns, euonymus leaves, and so on. Since Kuan Pinggong et al. reported that nitrogen-doped carbon nanotubes could improve activity in 2009 [13], nitrogen-doped carbon materials employed as metal-free catalysts have become one of the hot topics regarding fuel cell catalysts [14]. Nitrogen atoms can replace some carbon atoms in the carbon structure, resulting in structural distortion [15], the alteration of charge density, and the electronic structural change of carbon materials such that the free energy of adsorption of reactants and intermediate objects on the material is changed, thus further changing the ORR properties of carbon materials. A nitrogen source is indispensable in the preparation of nitrogen-doped, carbon-based catalysts and it can be used as the active site. Using biomass waste as a carbon source [16] and nitrogen-containing compounds such as urea, melamine, and hydrazine hydrate as a nitrogen source [17], nitrogen atoms are doped in carbon nanomaterials after hydrothermal and high-temperature treatment [18].

The interior of a walnut shell has a developed pore structure [19]. Its molecular structure contains hydroxyl [19], carbonyl [20], carboxyl [21], hydroxyl [22], aryl [23], amino [24], conjugate double bond, and other active groups [25]. In recent years, it has been popularly used as biochar produced from agricultural and forestry-based waste [26]. Walnut shell mainly consists of cellulose, accounting for 30.88% of the total content; hemicellulose [27], accounting for 27.26% of the total content; and lignin, accounting for 38.05% of the total content [28]. It presents a layered, porous structure at the microscopic level [29]. In addition, the annual mass planting of walnuts in Shanxi produces walnut shell waste [30], which is a great burden to society and a waste of energy [31]. In the present study, we used walnut shell as the precursor of a biomass carbon material and prepared an efficient porous ORR catalyst via simple hydrothermal and high-temperature annealing [32].

## 2. Experimental Procedure

### 2.1. Material

Walnut shells purchased with permission from Shanxi were used as precursors. All chemicals mentioned in this article were purchased from licensed manufacturers and can be used without any further purification.

### 2.2. Material Synthesis

Walnut shells used in the current study were produced in Shanxi Province. The inner pulp was removed, and the hard shell was crushed into a particle size of 75 microns with a grinder. Then, the walnut shell powder was ultrasonically cleaned for 60 min with deionized water and dried at 80 °C overnight; finally, yellow walnut shell powder (HTKC) was obtained. Then, the urea was deposited into a crucible, which was covered by the crucible’s lid; placed into a tubular furnace; heated to 550 °C at a 5 °C/min heating rate; and stored in nitrogen atmosphere for 4 h. Finally, light-yellow g-$C_{3}N_{4}$ is obtained.

The walnut shell was mixed with g-$C_{3}N_{4}$ at a ratio of 1:5, ground, poured into 30 mL of deionized water, and placed into an ultrasonic instrument for 30 min to obtain a uniform solution. The mixed solution was poured into a reaction kettle and heated in an oven at 180 °C for 15 h. The sample was washed and dried. Under a nitrogen atmosphere, the temperature was raised to 900 °C at a rate of 5 °C/min in a tubular furnace, and this temperature was held for 4 h. Then, after the sample was cooled to room temperature, the black solid powder was removed, and the sample was ground, pickled, placed in neutral pH water, and washed once with ethanol. After drying at 80 °C in the oven, a black powder was obtained and denoted as NSCL-900. For comparison, a non-urea-doped sample was prepared and labeled NNS-900. Urea was directly deposited into a reaction kettle with walnut shell without pyrolysis and labeled NS-900.

### 2.3. Physical Characterization

The crystal structure and carbonization degree of the catalysts were observed at 0° to 90° at 5°/min on an X-ray diffractometer (XRD, Ultima IV X-ray, Rigaku, Cu Ka). Scanning electron microscope (SEM, Hitachi Regulus8100) and transmission electron microscope (TEM, TALos F200X) were mainly used to observe the microstructures of catalysts under high magnification. A microscopic Raman spectrometer (HORIBA Xplora Plus, 532 nm laser excitation), the determination of specific surface area (SSA), and pore structure analyzer (Micromeritics, ASAP2020) were used to characterize the composition of the samples and the disorder and defect changes of the catalysts’ structure. SSA was determined by the Brunauer–Emmett–Teller (BET) method. X-ray diffractometer (XPS, Model: Phi-5000 versaprobeiII) was used to analyze the elemental composition and content of the catalysts’ surface.

### 2.4. Electrochemical Measurement

#### 2.4.1. Measuring Devices

The ORR performance of the prepared catalysts was measured using the Chenhua CHI760E electrochemical workstation. Measurements were made at room temperature using a three-electrode system. A glassy carbon electrode (GC, d = 5 mm) coated with catalyst ink was used as the working electrode, Ag/AgCl (3 M KCl) was used as the reference electrode, and graphite rod was used as the counter electrode.

#### 2.4.2. Preparation of Catalyst Ink

The catalyst was thoroughly ground. Then, 750 μL of deionized water, 250 μL of isopropyl alcohol, and 20 μL of Nafion solution were added to 5 mg of catalyst powder. They were placed in a 1.5 mL centrifuge tube and treated with ultrasound for 1 h so that they were fully mixed. A pipette gun was used to measure 10 μL of catalyst ink and drop it on the glassy carbon electrode, which was left to dry naturally (the average load was $0.25\;\mathrm{mg}\;{\mathrm{cm}}^{-2}$).

#### 2.4.3. ORR Performance Test

Cyclic voltammetry curves (CV) were determined in 0.1 mol KOH electrolyte in oxygen and nitrogen atmospheres. The test range was −0.8~0.2 V and the scanning rate was 50 mV/s. Linear sweep voltammetry (LSV) was applied to a rotating disk electrode (RDE) at the speed of 10 mV/s under the conditions of 400~2500 rpm, and the test range was −0.8~2 V. For convenience during calculation, all voltages were standardized with reversible hydrogen electrodes. According to the RDE data, the slope of the linear fitting line was used to calculate the electron transfer number (*n*) according to the Koutecky–Levich (K–L) equation:(1)$$\frac{1}{j}=\frac{1}{j_{L}}+\frac{1}{j_{K}}=\frac{1}{B\omega^{1/2}}+\frac{1}{j_{K}}$$
(2)$$B=0.62nFC_{0}{\left(D_{0}\right)}^{2/3}\upsilon^{-1/6}$$

In this equation, *j* is the measured current density; $j_{K}$ and $j_{L}$ are the kinetic and diffusion limit current densities, respectively; ω is the rotational angular velocity of the electrode; n is the electron transfer number; F is Faraday constant ($96,482\;{\mathrm{mol}}^{-1}$); $C_{0}$ is the saturation concentration of O_2_ in 0.1 M KOH solution at room temperature ($1.2\times 10^{-6}\;\mathrm{mol}\;{\mathrm{cm}}^{-3}$); and υ is the kinematic viscosity of the electrolyte at room temperature ($0.1\times {\mathrm{cm}}^{2}\;s^{-2}$). According to the K–L diagram, the slope (1/B) can be used to calculate the number of electron transfers. Electron transfer number (n) and peroxide percentage ($H_{2}O_{2}\%$) are calculated based on the following equation:(3)$$n=\frac{4I_{d}}{I_{d}+I_{r}/N}$$
(4)$$H_{2}O_{2}\%=\frac{200I_{r}/N}{I_{d}+I_{r}/N}$$

In this equation, $I_{d}$ is the disk current density, $I_{r}$ is the ring current density, and N is the current collection efficiency of $P_{t}$ (0.37).

For the stability and methanol toxicity tests of the catalyst, the stability of the prepared catalyst and commercial P_t_/C was tested at a potential of 0.8 V vs. RHE in an O_2_-saturated 0.1 M KOH electrolyte at a speed of 1600 RPM for 2 h. The test conditions for methanol toxicity were the same as above, and i-*t* test was also adopted. The test time was 500 s and methanol was added at the 300th s at a concentration of 1 M.

## 3. Results and Discussion

### 3.1. Morphology Characterization of Catalyst

As shown in Figure 1, the SEM images display the morphological characteristics and microstructural changes of the prepared catalyst surface. Figure 1a,b are the SEM scanning results concerning NSCL-900. The carbon structure formed by urea doping after pyrolysis is a carbon nanosheet with an increased pore structure and abundant defects. Figure 1c,d are SEM images of NS-900. Although urea doping also formed defects and pore structures, they were far inferior to those of NSCL-900. Figure 1e,f show NS-900. The catalyst prepared without urea doping has a smooth surface; it does not have a pore structure, nor does it possess any defects. It can be seen that urea promotes an increase in the number of catalyst pores and defects and that the pyrolysis effect of urea is superior. The increase in the pore structure and number of defects of the catalyst provides active sites inside the catalyst, thus improving its catalytic activity.

### 3.2. Structural Characterization of Catalyst

The crystal structures of NSCL-900, NS-900, and NNS-900 were analyzed by XRD (Figure 2a). The diffraction peaks of NSCL-900, NS-900, and NNS-900 are centered on $2\theta =24.0^{{}^{\circ}}$ and $43.0^{{}^{\circ}}$ [33]. In addition, they can be attributed to the 002 and 101 lattice planes of typical amorphous carbons [34]. Compared with NNS-900, the half peak of the diffraction peak of NS-900 widens and the peak height shortens, indicating that the introduction of a nitrogen atom causes a change in the carbon crystal structure and reduces the graphitization degree [35]. Further analysis and comparison between NS-900 and NSCL-900 shows that the peak height of NSCL-900 is lower, indicating that the g-CN molecular group formed after the pyrolysis of urea at 550 °C is more likely to break carbon chains in a hydrothermal environment, and more nitrogen atoms are introduced, resulting in a lower graphitization degree [36]. This result is consistent with that obtained via Raman spectroscopy.

Raman spectroscopy effectively revealed the defect levels and graphitization degrees of NSCL-900, NS-900, and NNS-900 [37]. As shown in Figure 2b, peaks D and G can be observed at $1350{\;\mathrm{cm}}^{-1}$ and $1590{\;\mathrm{cm}}^{-1}$. Peak D is generally regarded as the disordered vibration peak of graphene, which is used to characterize the defects of carbon-based catalysts [38]. Peak G is the main characteristic peak of graphene, reflecting the number of layers of graphene. $I_{D}/I_{G}$ is an important index of the defect level and graphitization degree of a catalyst [39]. According to spectral calculation, the $I_{D}/I_{G}$ values of NSCL-900, NS-900, and NNS-900 are 1.05, 0.93, and 0.86, respectively, indicating that the introduction of urea will cause the disorder of the carbon structure and that the urea treatment will increase the defect level of the carbon structure [40].

XPS analysis was performed on NS-900, NNS-900, and NSCL-900 to mainly analyze their surface composition and chemical bonds [41]. Figure 3a shows the total spectrum analysis of NS-900, NS-900S and NSCL-900. It was found that the spectra of NS-900 and NSCL-900 all show three peaks. The peak centers are 285, 400, and 532 eV, respectively, corresponding to C1s, N1S, and O1s [42]. However, there are only two peaks shown in the spectrum of NNS-900. The peak centers are 285 and 532 eV, corresponding to C1s and O1s [43], indicating that the walnut shell, which is itself a carbon base, does not produce nitrogen atoms, and that the nitrogen atoms are introduced by urea. By analyzing the total spectrum content (Figure 3b), it was found that the total nitrogen fractions of NS-900 and NSCL-900 are 3.58% and 9.57%, respectively, indicating that urea pyrolysis can affect the total nitrogen content of hybrid carbon materials. This is because the hydrogen and oxygen atoms volatilize after the urea treatment, leaving behind a carbon–nitrogen polymer, and subsequently reducing the involvement of other heteroatoms with nitrogen atoms. In order to further analyze the bonding of the elements, we analyzed the high-resolution XPS spectra of C1s and N1s of the catalyst.

Figure 4a shows the high-resolution C1s spectra of NSCL-900. There are four characteristic peaks located at 284.7 eV, 285.9 eV, 287.3 eV, and 289.2 eV, corresponding to the C-C /C=C bond, C–N/C=N bond, C=O bond, and O–C=O bond, respectively [44]. In Appendix A, NS-900, NSCL-900, and NS-900 show high-resolution C1s spectra. Figure 4c shows the high-resolution O1s spectrograms of NSCL-900. There are two characteristic peaks: one corresponds to the COO– bond and C=O bond at 531.6 eV; the other is the O–C bond at 533.eV [45]. The bonding confirms that N has been introduced into the carbon network structure.

Figure 4b shows the high-resolution N1s spectra of NS-900, NNS-900, and NSCL-900. There is no N in NNS-900, so the image does not have obvious characteristic peaks. However, NS-900 and NSCL-900 have four obvious characteristic peaks due to the doping of N, which are located in the range of 398.10–398.84, 399.54–399.89, 400.33–400.82, and 401.39–402.15 eV, corresponding to pyridin-N, yrro-N, raphite-N, and a nitrogen oxide bond [46]. In addition, we analyzed the peak area of nitrogen functional groups of NSCL-900 by peak fitting (Figure 4d) and found that pyridin-N, pyrrorole-N, graphite-N, and nitrogen oxide accounted for 30.9%, 35.2%, 29.2%, and 4.5% of the total nitrogen, respectively. In Appendix A, the Pyridine-N of NS-900 accounted for 28.94% of the total nitrogen, pyrrole-N accounted for 32.15%, and graphite-N and oxide-N accounted for 29.90% and 8.68%, respectively. NNS-900 has no nitrogen functional groups because it was undoped with nitrogen. The nitrogen functional groups of NS-900 and NNS-900 showed that the nitrogen functional groups of these two catalysts were not as high as those of NSCL-900. According to previous experimental studies, pyridine-N and pyrrorole-N can increase the number of active sites of ORR and graphite-N can increase the limiting current density of the catalyst. The three nitrogen functional groups of the prepared NSCL-900 accounted for 95.3% of the total nitrogen, indicating that the catalyst has high catalytic activity. Moreover, the N content of NSCL-900 is also higher than that of the other two samples, proving that urea pyrolysis is more beneficial for the formation of pyridine-N, pyrro-N, and graphite-N [47].

### 3.3. Electrochemical Performance of Catalyst

In order to determine the ORR performance of the catalyst, the ink of NSCL-900, NS-900, and NNS-900 were dried on the surface of a GC-RDE electrode and further tested by cyclic voltammetry (CV) and linear sweep voltammetry (LSV) in solution under a saturated oxygen atmosphere. Figure 5a shows the CV curves of NSCL-900, NS-900, and NNS-900. It can be seen from the figure that in the oxygen atmosphere, there are obvious oxygen reduction peaks, indicating that all the catalysts have ORR activity, and the potential sequence of oxygen reduction peak is NNS-900 (0.65) < NS-900 (0.70) < NSCL-900 (0.78). In order to further explore the ORR performance of the NSCL-900, NS-900, and NNS-900 catalysts, we tested the LSV curves under the condition of a 1600 rmp electrode rotation speed, and the results are shown in Figure 5b. The initial potential of NS-900 is 0.83 V vs. RHE. The initial potential of NNS-900 is 0.75 V vs. RHE. The initial potential of NSCL-900 is 1.00 V vs. RHE, which is higher than the initial potential of commercial P_t_/C (20wt%) (0.92 V vs. RHE). In addition, the half-wave potential ($E_{1/2}$) shown by NSCL-900 is 0.86 V vs. RHE, which is significantly higher than the half-wave potential ($E_{1/2}$) of NS-900 (0.72 V vs. RHE), the half-wave potential ($E_{1/2}$) of NNS-900, and the commercial half-wave potential of P_t_/C (20wt%) $\left(0.83\;V\;\mathrm{vs}.\;\mathrm{RHE}\right)$. In addition, NSCL-900 exhibits higher current-limiting density than NS-900 and NNS-900. These results confirm the excellent ORR activity of NSCL-900. This excellent ORR activity is attributed to the fact that g-CN molecular groups formed by pre-treated urea increase the level of nitrogen doping of the catalyst during the pyrolysis process and cause the formation of a high content of pyridine nitrogen and pyrrole nitrogen [48].

Furthermore, Tafel analysis was used to obtain kinetic information. As shown in Figure 5g, the Tafel slope ($72{\;\mathrm{mV}\;\mathrm{dec}}^{-1}$) of NSCL-900 is close to the reference Pt/Cde Tafel slope ($72{\;\mathrm{mV}\;\mathrm{dec}}^{-1}$), indicating that there is a high exchange current density at the interface between the catalyst and the electrode, which is most favorable for ORR catalytic activity. However, the Tafel slope ($172{\;\mathrm{mV}\;\mathrm{dec}}^{-1}$) and Tafel slope ($122{\;\mathrm{mV}\;\mathrm{dec}}^{-1}$) of NNS-900 indicate that the ORR dynamic process is poor. These results indicate that after pretreatment of urea, nitrogen atoms are more likely to break chemical bonds between carbon atoms and form pyrrole nitrogen and pyridine nitrogen at the same pyrolysis temperature and time. As confirmed by XPS, NSCL-900 shows superior ORR performance at initial potential and limiting current density [49].

In order to further verify the ORR reaction mechanism, the catalyst was coated on a rotating ring disk electrode (RRDE) in a saturated 0.1 M KOH solution at a rotation speed of 400 to 2050 RPM. The limiting current density increased with the increase in the rotation speed, and the diffusion distance decreased with the increase in the rotation speed The measurement results are shown in Figure 5d. Both NSCL-900 and 20wt% P_t_/C show low ring current density, indicating that less ring current is detected on the ring electrode, and the results indicate that the catalyst has high catalytic activity. As shown in Figure 5d, we calculated electron transfer (*n*) and $H_{2}O_{2}$ yield using a formula based on RRDE ring current and disk current data. The n of NSCL-900 ranges from 3.90 to 3.98 and the $H_{2}O_{2}$ yield is lower than 6.5% in the potential range of 0.2 to 1.0 V, which is slightly higher than that of Pt/C (0.67%), which proves that the ORR reaction of NSCL-900 is in accordance with the four-electron reaction. This is consistent with the calculation results obtained using the K–L equation (Figure 5f).

To further investigate the stability of the catalyst, we tested NSCL-900 and 20wt% P_t_/C via current-time timing measurements (Figure 5h). The saturated RDE was continuously rotated at 1600 rpm in the solution and it was found that, after the 1500 s test, NSCL-900 and Pt/C could still reach 78.3% and 72.5% of the initial current. The results show that the catalyst NSCL-900 has better long-term stability than 20wt% P_t_/C.

The methanol-poisoning resistance test is also an important indicator of the stability of an ORR catalyst. As shown in Figure 5i, in an oxygen-saturated solution of 0.1 m KOH, 3 mol methanol was dropped at 600 s to observe the changes in the cathode current of NSCL-900 and 20wt% P_t_/C according to the i–t curve. The results show that the catalyst NSCL-900 reached 79.3% of the initial current after 2000 s, while 20wt% P_t_/C only reached 70.1% of the initial current, which clearly indicates that the prepared catalyst NSCL-900 has a higher ability to resist methanol poisoning than P_t_/C. Compared with 20wt% P_t_/C, NSCL-900 has better stability and methanol toxicity resistance and is a metal-free biochar ORR catalyst with great developmental potential and application prospects.

In summary, the high catalytic activity of NSCL-900 can be attributed to two aspects. On the one hand, its abundant mesoporous structures can improve electron transfer efficiency and expose a large number of active sites; this helps accelerate the reaction. On the other hand, due to the addition of nitrogen elements, the functional groups (pyridine nitrogen and pyrrole nitrogen) on the surface of the carbon matrix are regulated, and the catalytic activity of the ORR is improved.

## 4. Conclusions

Using walnut shell as a biomass to prepare and develop high-performance carbon-based catalytic materials can not only reduce the impact of agricultural waste on the environment but also solve the problems of high costs and a limited supply of platinum and other precious metal catalysts. The reasonable utilization and conversion of biomass resources can not only solve the problem of the lack of fossil resources but also realize the effective utilization of biomass resources and their waste. The results show that biomass offers great potential with respect to its use as a carbon-based material for the cathodes of fuel cells. The main purpose of this study is to activate walnut shell biomass and find a low-cost method with which to convert waste into a high-value product. This study provides a novel and simple method to prepare high-performance ORR catalysts.

In this work, a commercially available and low-cost method was reported for the manufacture of porous, nitrogen-containing carbon materials using walnut shells. The synthesis process consists of an initial pretreatment of urea, followed by the doping of the treated urea with walnut shell powder for hydrothermal annealing, and finally annealing in a nitrogen atmosphere. The surface morphology and structure of the catalyst were characterized by SEM, TEM, Raman spectroscopy, and XPS, and the mechanism of the catalyst’s activity was further explained. The results showed that the nitrogen content of the pretreated urea was higher than that of untreated urea. The doping and activation methods showed high-performance electrochemical activation, excellent tolerance to methanol, and resistance to carbon monoxide toxicity. This method provides a feasible, mimicable, and scalable method for the preparation of high-performance ORR catalysts from rarely used biomass waste.

In summary, we describe a low-cost, simple, and easy-to-manufacture method. Using walnut shell biomass waste as a carbon base, pyrolysis urea was successfully synthesized into a highly nitrogen-doped, high-activity ORR catalyst. The initial potential of the catalyst was 1.00 V vs. RHE, while the half-wave potential ($E_{1/2}$) of the prepared catalyst was 0.86 V vs. RHE. The catalyst’s excellent catalytic activity was attributed to the high content of pyridine nitrogen, pyrrole nitrogen, and graphite nitrogen. This research shows that the doping effect is poor when urea is not treated, and that the nitrogen content is only 3.58%. After urea pyrolysis, the nitrogen content is as high as 9.57% under the same conditions. The current work explores a new method for the use of low-cost nitrogen doping, which is expected to provide a theoretical basis for the subsequent preparation of superior ORR catalysts [6].

## Figures and Tables

**Figure 1 molecules-28-02072-f001:**
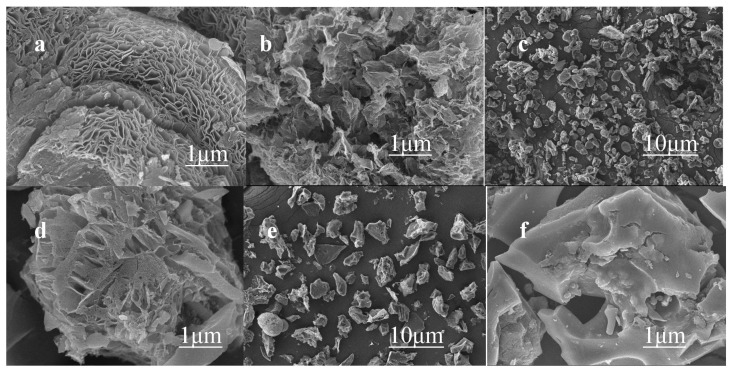
SEM images of NSCL-900 (**a**,**b**), NS-900 (**c**,**d**), and NNS-900 (**e**,**f**).

**Figure 2 molecules-28-02072-f002:**
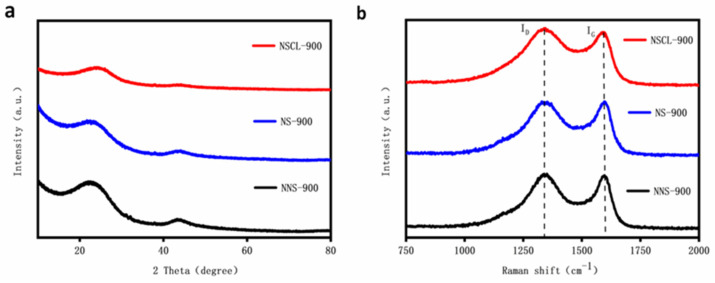
XRD patterns (**a**) and Raman Spectra (**b**) of NSCL-900, NS-900, and NNS-900.

**Figure 3 molecules-28-02072-f003:**
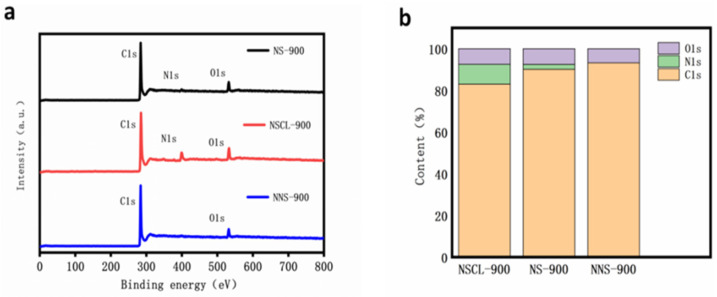
XPS spectrograms of NS-900, NSCL-900, and NNS-900: (**a**) illustration of content of various elements in NSCL-900, NS-900, and NNS-900 (**b**).

**Figure 4 molecules-28-02072-f004:**
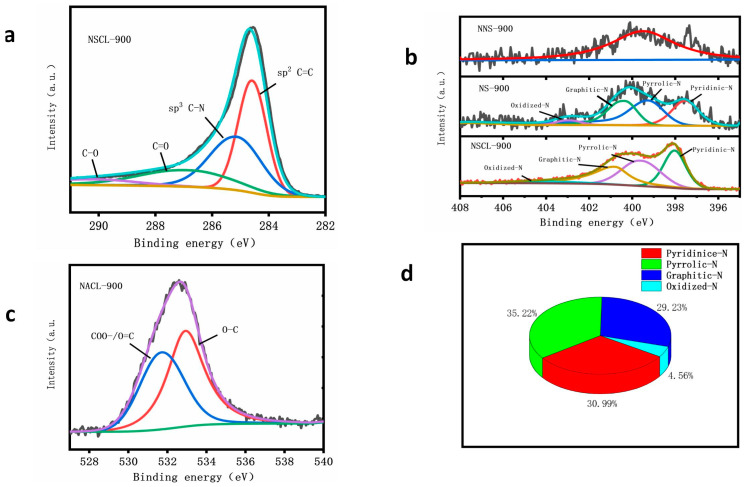
High-resolution N1s spectrograms of NSCL-900, NS-900, and NNS-900 (**b**), and high-resolution C1s (**a**) and O1s (**c**) spectrograms of NSCL-900. Figure (**d**) shows the content of nitrogen functional groups in NSCL-900.

**Figure 5 molecules-28-02072-f005:**
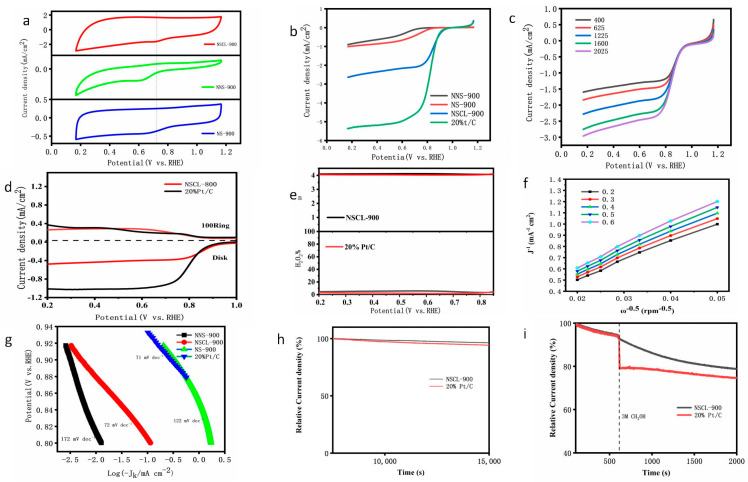
CV curves of NSCL-900, NNS-900, and NS-900 in 0.1 M KOH solution in a saturated atmosphere at room temperature and a scanning rate of 50 mV/s (**a**). LSV curve of NSCL-900 at electrode rotation rate of 1600 rpm and scanning rate of 10 mV/s (**b**). LSV curves of NSCL-900 at different speeds of 400-2025 RPM (**c**) correspond to K–L diagrams (**f**) of NSCL-900 at 0.2 V to 0.6 V, respectively. RRDE linear sweep voltammetry (**d**) of NSCL-900 and 20%Pt/C in a saturated 0.1 M KOH solution at a motor rotation rate of 1600 rmp and a sweep rate of 5 mV/s. The electron transfer number *n* (up (**e**)) and yield (down E) were calculated from the RRDE measurements of NSCL-900 and 20%Pt/C. General Tafel slope curve (**g**). Stability curves (**h**) for 15,000 s of NSCL-900 and 20%P_t_/C in a saturated 0.1 M KOH solution at the electrode rotation rate of 1600 rmp, and toxicity curves (**i**) for methanol resistance determined by time-amperometric rule.

## Data Availability

Not applicable.

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
