# Peer review of "High-Efficiency Oxygen Reduction Reaction Revived from Walnut Shell"

_molecules, 2023, doi:10.3390/molecules28052072_

Round 1

Reviewer 1 Report

In the manuscript, a highly efficient porous ORR catalyst was prepared by simple hydrothermal and high temperature annealing using walnut shell as the precursor of biomass carbon material. The initial potential of NSCL-900 is 1.00 V vs. RHE and the half-wave potential increases is 0.85 V vs. RHE, showing catalytic activity superior to 20 wt% Pt/C. Therefore, this work provides a good theoretical basis for exploring a new way to prepare non-metal ORR catalysts with high activity and low cost. I strongly recommend that this paper is accepted and published.

In addition, the author needs to add some more in-depth discussion and a more detailed explanation of the results.

1. It is mentioned in Figure 1 that the urea after pyrolysis can effectively promote the increase of catalyst pores and defects, but the relationship between this conclusion and catalytic activity is not explained.

2. The high-resolution C 1s and O 1s spectra of NNS-900 and NS-900 are not involved in the Figure 4 , which is lack of contrast. In addition, Figure 4d lack the content of nitrogen functional groups of the contrast sample NS-900.

3. Repair the format of Figure 1, Figure 4 and Figure 5.

Reviewer 2 Report

This manuscript described a new method of preparing non-metal ORR catalyst with walnut shell as precursor and g-C3N4 as nitrogen source by hydrothermal and high temperature annealing. The catalyst NSCL-900 showed excellent catalytic activity and stability. Therefore, walnut shell has potential application prospects in the field of non-metallic ORR catalysts. However, there are still some deficiencies that need to be modified or further explained in some chapters.

1. How to deal with walnut shell waste in the past? What are the advantages of the new way of preparing non-metal ORR catalyst described in the manuscript?

2. In Figure 2a, which diffraction peak change shows that the introduction of nitrogen atoms causes the change of carbon crystal structure and reduces the degree of graphitization?

3. Some important refs may be cited to enrich reference part of the revised manuscript, such as, Electrochem. Energy Rev. 2022, 5, 7; Electrochem. Energy Rev. 2022, 5, 32.

4. It is mentioned in the article that the half wave potential of NSCL-900 is 0.85 V vs. RHE, but according to Figure 5b, the value is higher, so it is recommended to re analyze the experimental data.

5. The name of catalysts should be consistent in the article.
